# Psychological Well-Being of Left-Behind Children in China: Text Mining of the Social Media Website Zhihu

**DOI:** 10.3390/ijerph19042127

**Published:** 2022-02-14

**Authors:** Yuwen Lyu, Julian Chun-Chung Chow, Ji-Jen Hwang, Zhi Li, Cheng Ren, Jungui Xie

**Affiliations:** 1School of Economics and Statistics, Guangzhou University, Guangzhou 511400, China; 1111764007@e.gzhu.edu.cn; 2Center for Information Technology Research in the Interest of Society and the Banatao Institute, University of California, Berkeley, CA 94720, USA; 3School of Social Welfare, University of California, Berkeley, CA 94720, USA; jchow99@berkeley.edu (J.C.-C.C.); cheng.ren@berkeley.edu (C.R.); 4Center for General Education, Chung Yuan Christian University, Taoyuan 320314, Taiwan; jhwang29@gmu.edu; 5School of Policy and Government, George Mason University, Arlington, TX 22030, USA; 6School of Information, University of California, Berkeley, CA 94720, USA; zhili@berkeley.edu; 7School of Public Administration, Guangzhou University, Guangzhou 511400, China

**Keywords:** left-behind children, personal narrative text, text mining, linguistic analysis, textual analysis, psychological well-being

## Abstract

China’s migrant population has significantly contributed to its economic growth; however, the impact on the well-being of left-behind children (LBC) has become a serious public health problem. Text mining is an effective tool for identifying people’s mental state, and is therefore beneficial in exploring the psychological mindset of LBC. Traditional data collection methods, which use questionnaires and standardized scales, are limited by their sample sizes. In this study, we created a computational application to quantitively collect personal narrative texts posted by LBC on Zhihu, which is a Chinese question-and-answer online community website; 1475 personal narrative texts posted by LBC were gathered. We used four types of words, i.e., first-person singular pronouns, negative words, past tense verbs, and death-related words, all of which have been associated with depression and suicidal ideations in the Chinese Linguistic Inquiry Word Count (CLIWC) dictionary. We conducted vocabulary statistics on the personal narrative texts of LBC, and bilateral *t*-tests, with a control group, to analyze the psychological well-being of LBC. The results showed that the proportion of words related to depression and suicidal ideations in the texts of LBC was significantly higher than in the control group. The differences, with respect to the four word types (i.e., first-person singular pronouns, negative words, past tense verbs, and death-related words), were 5.37, 2.99, 2.65, and 2.00 times, respectively, suggesting that LBC are at a higher risk of depression and suicide than their counterparts. By sorting the texts of LBC, this research also found that child neglect is a main contributing factor to psychological difficulties of LBC. Furthermore, mental health problems and the risk of suicide in vulnerable groups, such as LBC, is a global public health issue, as well as an important research topic in the era of digital public health. Through a linguistic analysis, the results of this study confirmed that the experiences of left-behind children negatively impact their mental health. The present findings suggest that it is vital for the public and nonprofit sectors to establish online suicide prevention and intervention systems to improve the well-being of LBC through digital technology.

## 1. Introduction

During China’s planned economy period, the movement of citizens was controlled, and citizens were allowed to reside only in their registered permanent residence, according to the household registration (Hukou) system. Since the economic reform of 1978, China has worked towards a market economy and removed barriers deterring mobility, which resulted in the migration of tens of thousands of residents from rural areas to urban ones [1]. According to the latest report from the government, the migrant population in China has reached 290 million, an increase of 11.89% as compared with the 2010 Chinese Census [2]. On the one hand, migrant workers have tremendously contributed to China’s economy; on the other hand, the children of migrant workers are facing vital problems that need urgent attention [3]. Left-behind children (LBC) are children younger than 17 years old who stayed in the family’s registered permanent residence while their parent or parents migrated to another area for work; LBC need attention and care from the government and society [4]. The 2015 China Family Development Report indicated that LBC in rural areas accounted for 35.1% of the total population of children [5]. According to the National Information Management System developed by the Ministry of Civil Affairs, regarding LBC and children in difficulty in rural areas, there were 6.97 million LBC in 2018, a 22.7% decrease from the 9.02 million LBC recorded in 2016 [6]. However, researchers have taken the Sixth National Population Census (2010) of the People’s Republic of China as sample data, and have estimated the number of LBC to be 61.02 million [7]. Irrespective of the method used, i.e., a recorded number or an estimated number, the population of LBC is significant, and their education and psychological development is an urgent social problem for the Chinese government.

In a recent study, researchers reported that male LBC were at higher risk of skipping breakfast, physical inactivity, internet addiction, having ever smoked tobacco, and suicidal ideation, whereas female LBC preferred sugary drinks and had higher risks of alcohol addiction and suicidal ideations than their non-LBC counterparts [8]. In addition, Z. Jia and Tian found that LBC were 2.5 times more likely to suffer from loneliness, and 6.4 times more likely to be very lonely [9]. In another study, conducted using a cross-sectional design with 3254 participants (8 to 17 years old), the results showed that the prevalence of depression was 10.9% among non-LBC and 14.3% among LBC [10]. From the data released in the 2011 Survey on Social Integration of Migrant Children in Wuhan, the results showed that LBC and migrant children had worse psychological and behavioral outcomes than local children, which disappeared with family and school support; this implied that family intervention and education policy reform could improve the psychological and behavioral outcomes of migrant children and LBC through family intervention [11].

However, existing research on LBC mostly consists of data collected and analyzed through questionnaires, mental health standardized scales, and traditional interview methods. Few researchers have focused on big data analytics of social media data. In the current digital age, social network services have become a part of everyone’s daily lives; texts, pictures, speech, and videos on social media all reflect the characteristics of people’s daily behaviors. As such, big data collected from social media can be very valuable in psychological studies, and therefore offers an important complement to traditional research methods [12]. In addition, text mining can be applied to assess large amounts of textual data that are continuously produced in social networks, by analyzing and discovering high-quality information from texts, to help analysts in decision making [13]. Recent research has applied text mining to the study of children’s social welfare. For example, Lyu et al. applied text mining techniques to analyze textual data collected from a social media platform, i.e., the Weibo website [14]. They investigated people’s attitudes toward child abuse incidents, as well as the reasons behind the abuse, and provided recommendations to the government to improve child protection policies. S. Li et al. conducted a study on 355 migrant youths in a middle school in Shenzhen [15]. They used a text mining technique, namely natural language analysis, on the children’s diary entries to examine the relationship between their language patterns and psychological resilience. Machine learning features were also applied to estimate their level of psychological resilience.

Since 1979, researchers have come to realize that the vocabulary used in oral communication can reflect personal feelings and thoughts. Therefore, a linguistic analysis can be used as an effective tool to understand a person’s psychological state [16]. In 1980, researchers discovered that patients who wrote about their emotions experienced improvements in their psychological well-being. The Linguistic Inquiry and Word Count (LIWC) was created because humans were not efficient or accurate enough when reading texts (LIWC 2015 dictionary categories can be seen in Table A1) [17]. The LIWC calculates psychologically meaningful words to determine a person’s psychological state. The dictionary has been widely used in linguistic analyses on topics such as depression, intimacy, and group psychology research [18]. A Chinese Linguistic Inquiry and Word Count (CLIWC) tool was developed by Taiwanese scholars, based on the unique categories of the Chinese language. The CLIWC has established reliability and validity, and can be used to detect mental health issues in Chinese texts with high accuracy [19].

Based on the psychological diagnosis results of linguistic analyses, an online suicide prevention and psychological intervention system could be established for LBC. Fang, Chang, and Fan extracted characteristic words related to mental disorders through text matching of depressive dairies published on social media, and then scored them [20]. When the score exceeded a certain level, the system would automatically send messages to the patient’s friends, alerting them that the patient might need psychological support. Parime and Suri used machine learning techniques to monitor online bullying on social media, in order to develop interventions that could reduce bullying [21]. Yang et al. used artificial intelligence technology to collect suicide-related keywords, such as “jumping off a building, cutting wrists, burning charcoal, and jumping into a river” to identify Weibo users with suicidal ideations, therefore helping to establish a suicide intervention mechanism [22]. Moreover, existing studies have mainly focused on depression, bipolar disorder, cyber bullying, and other groups, but few studies have focused on LBC. In this study, we identified LBC’s psychological problems through linguistic analysis and text mining by detected LBC with suicidal ideations, through their use of death-related words. The findings can be used to establish an online suicide prevention or psychological intervention for LBC in the future.

## 2. Methods

### 2.1. Data Source

In contrast to traditional data collection methods, such as questionnaires and standardized scales, in this study we created a computing application to quantitively collect personal narrative texts posted by LBC on Zhihu, a Chinese question-and-answer community website. Zhihu is similar to Quora in the United States, with two functions; namely, social networking and question-and-answer (Q&A) [23]. As of November 2018, Zhihu had more than 220 million users, posting more than 30 million questions and more than 130 million answers [24]. As China’s largest user content production platform, Zhihu contains user information on various groups in its massive text answers [25].

In this study, we used personal narrative texts under the three most popular questions (according to the views) posted about LBC on the Zhihu website as the research text data, with a total of 1475 personal narrative texts (1,013,829 words). Table 1 list the three most popular questions posted about LBC on the Zhihu.

Regarding each of the three questions, the occupation of the questioner, time the question was posted, and degree of attention it received are summarized in Table 2. In order to comply with the data privacy rights of the questioner, the ID name is represented by a code.

The details of the three questions about LBC are shown in Table 3, including the number of answers, the time frame for answering the questions, and the gender of the responders.

### 2.2. Data Collection

In this study, a web crawler computing application was implemented to gather firsthand textual data by accessing the webpage data, through the designed code of computer programming languages R and Python, to obtain useful data from massive data [26,27]. The process was implemented using Python programming, as shown in Figure 1. Selenium WebDriver technology was used to crawl Zhihu webpages and obtain all LBC’s personal narrative texts, as mentioned above.

To avoid violating the privacy protection of social media users’ personal information, a user’s ID number and name must be desensitized [28]. Thus, the application only collected the text content, time, and background information of the Q&A. The text data collected above were retrieved on 15 April 2020, and were analyzed for the first time.

### 2.3. Data Analysis

There are multiple studies that have performed data analyses from various positions, such as specialists in programming, policy, and child psychology. The analyses are produced through the same, or very similar, interpretations of researchers, to avoid the possibility of bias and reassess the assumptions, as well as to better determine a point of view which shows the most objectivity overall. The research steps used in this study are described below.

#### 2.3.1. Linguistic Analysis

Because the text of this study was in Chinese, the latest version of the CLIWC dictionary (from 2015), which includes 9720 types of Chinese vocabulary, was used. As early as 40 years ago, physician Walter Weintraub found that the number of times a person used first-person singular pronouns was related to people’s depression [18]. Since then, other scholars have found that people with depression or suicidal ideations used more first-person singular pronouns, negative words, past tense verbs, and death-related words in their writing [29,30]. Based on previous studies, we hypothesized that LBC would use negative emotion-based words, first-person singular pronouns, past tense verbs, and death-related words, which, in turn, relate to a higher degree of depression and stronger suicidal ideations. Therefore, four types of vocabulary in the CLIWC dictionary were adopted in this study: first-person singular pronouns, negative words, past tense verbs, and death-related words. Then, 1475 personal narrative texts of LBC on Zhihu (featuring 1,013,829 words), and a control group, were analyzed. If the proportions of the four types of vocabulary in the LBC text were higher than those of the control group, bilateral *t*-tests were used to test statistical differences between the two groups. The steps of the analysis are as follows:

Propose the research hypothesis.

**Hypothesis** **0** **(H0).***In the personal narrative text of LBC and the control group, the proportion of the four types of vocabulary in the total number of words is the same*.

**Hypothesis** **1** **(H1).***In the personal narrative text of LBC and the control group, the proportion of the four types of vocabulary in the total number of words is different*.

b.Calculate the statistics.

The two samples of LBC and control group texts have a known sample mean and standard deviation, and approximate a normal distribution. Therefore, in this study, we used a bilateral *t*-test to determine whether the two-sample means were significantly different [31]. The equation of the independent-sample *t*-test was as follows [32]:(1)$$t=\frac{{\bar{X}}_{1}-{\bar{X}}_{2}}{\sqrt{\frac{\left(n_{1}-1\right)S_{1}^{2}+\left(n_{2}-1\right)S_{2}^{2}}{n_{1}+n_{2}-2}\left(\frac{1}{n_{1}}+\frac{1}{n_{2}}\right)}}$$
where ${\bar{X}}_{1}$ and ${\bar{X}}_{2}\;$are the means of the two samples, respectively$;\;n_{1}$ and $n_{2}\;$are the sizes of the two samples, respectively$;\mathrm{and}\;S_{1}^{2}$ and $S_{2}^{2}\;$are the variance of the two samples, respectively.

c.Determine whether statistically significant differences exist between the study group and the control group.

First, we determined the *p*-value by checking the critical value table. If *p* < 0.05, the type of vocabulary in the text of the LBC group and the control group is statistically significant, rejecting H_0_.

All algorithms and results for the linguistic analysis were implemented by the research team in Python programming.

#### 2.3.2. Textual Analysis

A textual analysis obtains information by understanding the language in a body of text [33]. It is a generalized qualitative analysis method and a mainstream form of research in social science [34]. It does not emphasize the calculation and quantitative analysis of text, but uncovers hidden viewpoints and truth through theory, thereby effectively resolving the oppression that marginalized groups face in traditional research design [35,36].

To present a picture of LBC’s experiences, as well as to explore their psychological statuses and contributing factors, in this study we retrieved representative LBC’s texts through a lexical matching-based model [37]. The key words used for data extraction are summarized in Table 4.

## 3. Results

### 3.1. The Linguistic Analysis of LBC

To examine the mental state of the LBC in this study, we counted four types of vocabulary associated with depression and suicide in the texts of LBC, namely first-person singular pronouns, negative words, past tense verbs, and words associated with death, by text mining, and then calculated the proportion of total words for each type of vocabulary. Then, the personal narrative texts under the topic, but unrelated to LBC, on Zhihu, were randomly extracted as the control group. The specific text data can be found in Table A2. The counting results can be seen in Table 5, and the histogram can be seen in Figure 2.

As can be seen in Table 5, LBC used the four types of vocabulary more than the control group. The proportion of first-person singular pronouns among LBC was 5.37 times that of the control group, followed by negative words (2.99 times), past tense verbs (2.65 times), and death-related words (2.00 times). On this basis, bilateral *t*-tests were performed on the two sets of texts to determine whether the differences were statistically significant. Table 6 shows that the proportions of the four types of vocabulary used between the LBC and control groups were significant.

### 3.2. The Textual Analysis of LBC

Following the linguistic analysis of personal narrative texts, we extracted five representative personal narrative texts of LBC related to child neglect, and five representative personal narrative texts of LBC related to depression and suicide, as shown in Table 7 and Table 8. 

## 4. Discussion

### 4.1. Risk of Depression and Suicide

The present findings show that the proportions of four types of vocabulary (first-person singular pronouns, negative words, past tense verbs, and death-related words) in LBC personal narrative texts were all significantly higher than those in the control group, especially the proportion of first-person singular pronouns (5.37 times as high). Brockmeyer et al. reported that frequent use of first-person singular pronouns was associated with self-focused attention, which is a cognitive bias highly related to depression [38]. In a meta-analysis study, Holtzman argued that the use of first-person singular pronouns is a linguistic feature of people with depression, and that there is a correlation between the two [39]. Increased use of first-person singular pronouns has been linked to social isolation and suicide [40]. Therefore, the high rate of first-person singular pronouns among LBC suggests obvious self-focused characteristics, an important sign of depression and suicidal ideations.

The rate of use by LBC of negative words was 2.99 times that of the control group. Studies have shown that people with anxiety and depression showed attentional biases toward negative words [41]. People with depression showed less pleasure and arousal relative to positive words, but increased arousal relative to negative words [42]. The proportion of LBC using past tense verbs was 2.65 times higher than the control group, consistent with the finding that people with depression pay more attention to past events [43]. This suggests an explanation for why depression is often regarded as “being trapped in the past”, and is also reflected in the high use of past tense verbs in written disclosures [44,45].

With respect to death-related words, LBC used these words at twice the rate of the control group. In an analysis of letters written in the 2 years before suicide, researchers found that the frequency of using death-related words increased significantly [46]. Based on the linguistic analysis results, negative words, past tense verbs, and words associated with death in the texts of LBC were significantly more prevalent than those of the control group.

The findings of this study showed that LBC tended to use more first-person singular pronouns, negative words, past tense verbs, and death-related words. This suggests that LBC are deeply stuck in their painful left-behind experience, and excessively focus on themselves and their pasts. It is even more alarming that they are emotionally negative, and that some of them have attempted suicide or have suicidal thoughts and ideations.

### 4.2. Child Neglect as a Source of Depression and Suicidal Thoughts among the LBC in this Study

As previously discussed, the LBC in this study showed significant characteristics of depression and suicidal ideations. This research incorporated textual analysis of LBC personal narrative texts to present a deeper picture of possible causes.

Recent research has indicated that the proportion and degree of neglect experienced by LBC were higher than that of other children, especially, in rural areas in Western China [47]. LBC can be more prone to mental trauma and traumatic experiences than their counterparts [48]. Analyzing LBC’s personal narrative texts indicated that their mental well-being could be traced back to various types of neglect in early childhood.

Child neglect is defined as not meeting the basic needs of children, including adequate supervision, health care, clothing, and housing, as well as other physical, emotional, social, educational, and safety needs [49]. Child neglect is the most common type of child maltreatment, and affects the development and well-being of children worldwide [50]. However, because the damage caused by neglect is difficult to detect, and quantitatively measure, it has received far less attention in the field than child abuse [51].

First, the type of neglect that is unavoidable for LBC is parental emotional neglect. LBC are separated from at least one of their parents during childhood. Parental emotional neglect is related to externalizing problem behaviors [52], and is also a major risk factor for psychopathology, including internal problems such as depression [53]. Second, LBC’s parents migrate from underdeveloped areas to developed cities and participate in China’s economic growth. Due to the limited income conditions of migrant workers, many of them cannot provide a promising life for their LBC. This contributes to a certain degree of neglect related to health care, clothing, food safety, education, social, and other aspects of LBC. For example, the incidence of infectious and chronic diseases among rural LBC is 20% higher than that of non-LBC [54]. Their sacrifices are also seen as the price of China’s economic prosperity [55]. Finally, there is a lack of adequate supervision in addition to physical, social, and educational neglect. Among LBC in rural China, the percentages of LBC who experience victimization and polyvictimization have been reported to be 27.5% and 10.94%, respectively, and the poverty of LBC has been positively correlated to polyvictimization [56]. In addition, due to the lack of caregiver supervision, and the closed and monotonous living environment in LBC’s hometown, they often experience social and educational neglect. Some of them are also subjected to physical abuse and sexual assault [57]. These findings clearly show that LBC encounter various types and levels of child neglect during childhood, which lead to a higher risk of depression, suicide [58], and psychological trauma [59].

### 4.3. Online Suicide Prevention and Positive Intervention for LBC

The results of this study indicate that more LBC have severe experiences of child neglect, and therefore are more prone to depression and suicide. As illustrated in Table 7, some individuals with left-behind experience have suicidal ideations, or have exhibited suicidal behaviors. Suicide is a serious global public health problem [60]. The latest number of left-behind children in China reached 290 million [2]; therefore, it is imperative to establish an online suicide prevention system for LBC.

Social media is one of the important channels for vulnerable groups to express their emotions, and research has shown that they are more willing to self-expose in a public channel than in a private space [22]. Wang, et al. indicated that self-presentation in an online environment contributed to the treatment of depression [61]. Online technology is also increasingly being used in health and psychological care [62]; therefore, an online LBC group could be established to share and discuss their left-behind experiences and provide support for each other. In addition, spiritual leaders could share their successful experiences to help other helpless LBC cope with depression, and to teach them how to develop the ability to think positively. Meanwhile, digital technology has enabled a new way of thinking; i.e., that services and treatment could be provided online [63]. For example, social workers could offer positive psychological interventions through Zhihu’s LBC community. Therefore, it is conceivable to establish an online psychological intervention system for LBC. The relevant public departments could consider digital transformation of social work practices to innovatively solve the country’s public health crisis. This exploratory study demonstrates an opportunity for digital help for LBC, and offers new solutions to LBC’s public health problems.

## 5. Conclusions

This study used the psychometric dictionary CLIWC to conduct text mining through linguistic analysis, combined with textual analysis, of personal narrative texts written by LBC, which were collected from a social media website, Zhihu, in mainland China. Three key findings, obtained via quantitative analysis, are presented for discussion. First, LBC show a higher risk of depression and suicidal ideations than other children. Second, the psychological trauma of LBC stems from the different types and degrees of neglect suffered in their childhood. Finally, text mining was utilized to extract death-related words to identify LBC with suicidal risk, making a case to establish an online suicide prevention and psychological intervention system. More importantly, it provides a forward-looking approach that can be used for future research on the mental health and well-being of LBC and other vulnerable populations.

Inevitably, there are some limitations to this study, primarily concerning how to determine the authenticity of content on social media. It would be beneficial to consider a mixed-methods design; for example, combining big data analytics with traditional data collection methods to ensure the quality, rigor, and authenticity of the data. Text mining technology could be used as a supplement and extension of traditional research methods. For example, researchers could select social media users who fit the research topic and invite them to conduct online focus groups. From the perspective of digital public health, text mining technology could be used as a digital technology to improve public health problems. In summary, the exploratory research work of this study has important practical significance. It should facilitate the practical application of digital technologies in addressing public health problems.

## Figures and Tables

**Figure 1 ijerph-19-02127-f001:**
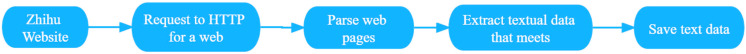
Basic principles of a web crawler.

**Figure 2 ijerph-19-02127-f002:**
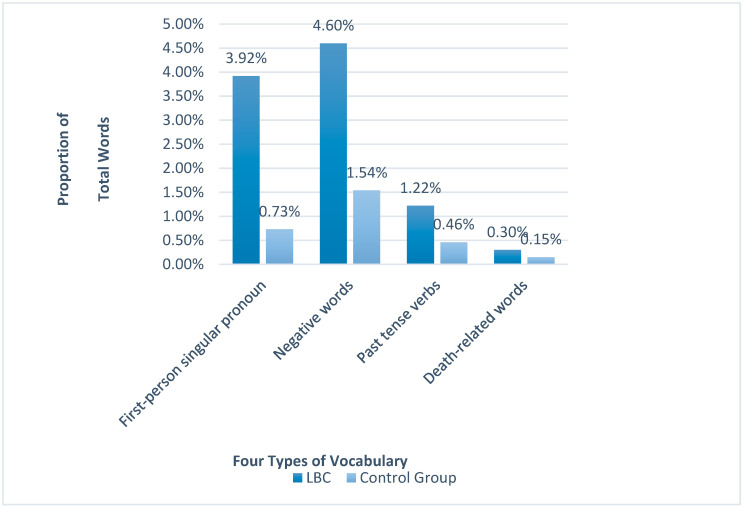
Histogram of four types of vocabulary between the LBC group and the control group.

**Table 1 ijerph-19-02127-t001:** The three most popular questions posted about LBC on the Zhihu website.

Questions about LBC on Zhihu
Q_1 What is the status of left-behind children when they grow up?
Q_2 What is the experience of left-behind children?
Q_3 What kind of psychological problems do left-behind children have when they grow up?

**Table 2 ijerph-19-02127-t002:** Basic information on the three studied questions posted on the Zhihu website.

Question	Questioner ID	Questioner’s Occupation	Time	Degree of Attention	
				Question Followers	Number of Views
Q_1	ID1	Art worker	25 July 2014	3936	1,403,306
Q_2	ID2	Parenting teacher	3 November 2015	1627	712,961
Q_3	Anonymous	Unknown	10 May 2014	551	334,302

**Table 3 ijerph-19-02127-t003:** Basic information on responses to the three questions about LBC.

Question	Number of Responses	Time Frame of Posted Answers	Gender of Responder		
			Male	Female	Unknown
Q_1	776	July 2014~December 2019	542	144	80
Q_2	561	November 2015~December 2019	345	137	79
Q_3	138	May 2015~January 2020	90	24	24

**Table 4 ijerph-19-02127-t004:** Key words for data extraction.

Lexical for Searching LBC’s Text	
Negative words	Get out, nightmares, quarrel, bullying, sexual assault, self-abasement, depression.
Death-related words	Suicide, taking drugs, die, despair, cut wrist.

**Table 5 ijerph-19-02127-t005:** Count results of four types of vocabulary between the LBC group and the control group.

Four Types of Vocabulary	LBC		Control Group	
	Counts	Proportion of Total Words	Counts	Proportion of Total Words
First-person singular pronoun	39,723	3.92%	1530	0.73%
Negative words	46,655	4.60%	3212	1.54%
Past tense verbs	12,351	1.22%	959	0.46%
Death-related words	3049	0.30%	311	0.15%
Total	101,778	10.40%	6012	2.88%

**Table 6 ijerph-19-02127-t006:** Bilateral *t*-test results of personal narrative text differences between the LBC group and the control group.

	First-Person Singular Pronouns	Negative Words	Past Tense Verbs	Death-Related Words
*t*	−20.0929	−15.8535	−14.0989	−9.65936
*p*-Value	2.76 × 10^−81^	5.90 × 10^−52^	2.19 × 10^−42^	3.36 × 10^−21^

*p* < 0.05.

**Table 7 ijerph-19-02127-t007:** Five representative personal narrative texts of LBC related to child neglect.

LBC_1: Every time I call [my parents], their first sentence will always be: “What happened?” I said nothing. “I just wanted to talk to you.” The rest of the conversation would be, they would answer whatever I asked, and then there would be no follow-up. Now when I call them, it’s mostly because I went out of money. It’s not often, once a month. It’s like having a period, which sometimes it’s early but sometimes late.
LBC_2: What I heard most since I was a child is, “Your parents work hard for you to make money, so you have to be obedient and don’t disappoint your parents.” “Your parents don’t want you anymore, so you need to live in someone else’s house.” “Get out of here, who wants your parents’ rotten money?” “Your parents have done a lot of evil in their previous life to have a child like you, just like a debt collector.” “If you hadn’t been born, your brother would have been the only child. How happy your parents would be.” … It’s really like a nightmare.
LBC_3: Although my parents sent letters and made phone calls from time to time, they never sent me a penny. All I remember is that my grandmother had a bad temper and used to quarrel and fight with my grandfather. Every day I took a lunch box and rice to the school dining room with steamed rice, and then I took some pickles and ate them. Occasionally, my grandmother gave me a dime to buy food at school (I vaguely remember the food provided by the school is 1-3 dimes). The child does not understand the hardships of life at all (my grandmother recalled that I hadn’t eaten any sugar for three years). Then such days continued until nearly 6th grade.
LBC_4: When I was bullied at school, other children had parents supporting them, but I didn’t, so I could only endure sexual abuse. I had nowhere to ask for help when I was sexually assaulted (let me be anonymous). I am good at studying. My relatives like me and my teachers also like me, but I am still an introverted and self-abasement person.
LBC_5: There are still many drawbacks since I was little without my parents around me. For example, I haven’t developed good study habits since I was young. There was very little homework in the countryside, so I could get first place casually in primary school. I have never studied seriously, and my grades were very unstable when I went to school in the city. Before I went to school in the city, I never knew that I needed to take notes in class, because no one took notes in my countryside school, and the teachers would not tell you to do so as well. I never studied English in the rural elementary school for six years (there are English classes, but you know that).

**Table 8 ijerph-19-02127-t008:** Five representative personal narrative texts of LBC related to depression and suicide.

LBC_6: When I first entered college, I felt low self-esteem so that I shed tears in the middle of the night. I like to put pressure on myself, hoping to make myself better. I don’t care about people in relationships and don’t know how to socialize with opposite-gender friends. Apart from genetic inheritance, the main reason for that is the experience of left-behind children. No matter what I can achieve in the future, I will not be happy inside.
LBC_7: Now I am inferior, sensitive, and insecure. I feel that my parents are terribly unfamiliar, and sometimes I feel embarrassed to stay with them. I hated them, why they gave birth to us but don’t care about us.
LBC_8: As a senior left-behind child, I lack of love, insecure, strong defensive heart, not talkative, and love to be alone.
LBC_9: I had severe depression five years ago. At that time, I had auditory hallucinations and committed suicide twice, and I am still taking medication. I was rescued by my parents. The first time I used a pair of tweezers in the toilet to stab my chest, and the second time I used underwear to strangle my neck in the hospital. Later, my mother called 120 (emergency) for me. I don’t know if everyone who wants to die has such despair as me. It just feels that in this world, there is really no love and no meaning in life. LBC_10: Hehe. I hate being with them, hate their words, hate their selfishness, hate their ignorance. At some point, I wish I had died suddenly.

## Data Availability

The data presented in this study are available on request from the corresponding author.

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
