# Peer review of "Psychological Well-Being of Left-Behind Children in China: Text Mining of the Social Media Website Zhihu"

_ijerph, 2022, doi:10.3390/ijerph19042127_

Round 1

Reviewer 1 Report

A more precise analysis of the communication dynamics that develop within the analyzed social network (Whizhu) is missing: who are asking? who responds? what are the questions? what is answered ?. There is also a lack of a more detailed and well-founded study of the profiles identified as LBC: age, gender, family economic situation, level of studies, place and neighborhood of residence, profession, possible police antecedents, etc. The authors must also explain what the criteria for choosing the texts chosen for the analysis have been before being analyzed.
In any case, after reviewing the indicated aspects, the text can be published.

Reviewer 2 Report

Text mining is a powerful data collection and analysis source used by governments and big corporations. The four vocabulary types are coherent with child care and health research.

On the other hand, if we study the situation of abandoned children worldwide, the results could be very similar. We would not have changed anything about his unfortunate emotional, physical and academic standing. Children from unstructured families have already been investigated and recognized this disadvantageous situation.

Text mining does not provide different information from the “traditional studies” based on questionnaires or interviews. On the other hand, the focus group can investigate specific situations in specific contexts, which is an excellent advantage over other data collection forms.

Criticism of traditional data collection systems lacks scientific rigour because it does not consider the quality of an investigation. It is necessary to consider fundamental research variables such as the purpose of the study, the research questions, the quality of the data collection instrument, the methodology employed, and finally, the novel knowledge and contributions to science.

This study has set a general objective that yields expected results and does not provide new knowledge. Therefore I suggest focusing the research on a specific aspect to be analyzed to look for alternatives, solutions and ways of improvement so as not to fall into an exercise of collecting information that is not transformed into concrete actions to improve or alleviate the situation of these poor children.

Reviewer 3 Report

This paper deals with a key issue in the area of mental health in China, through linguistic analysis and textual analysis of the personal narrative texts of left-behind children. Nevertheless, there are quite a number of aspects which I would suggest revising, namely: 

- I would suggest that the abstract in question would benefit from some form of framing of the context of the study: what knowledge does this paper add from an international perspective? how this/your study is increasing our understanding/knowledge? This would allow a better understanding of the importance of the topic. 

Introduction

- I did not feel that you adequately explained your rationale for choosing the variables which you studied, and so consequently I am unsure what theoretical impact this paper will have. Relevant literature is presented in support of the research problem, however, more actual references should be provided.

Methods

- There are no sources to help the reader understand the methodological approach (mixed-method social medial research) taken in the paper. This needs to be expanded, clarified, and supported by in-text citations.

- When were the data collected? Data collection period?

- Please outline the key ethical issues that you have taken in the management of Big Data (e.g. data protection; anonymity; privacy; re-use and publication)

- The process of analysis should be made as transparent as possible. The researcher’s own position should clearly be stated. For example, how they examined their own role, possible bias, and influence on the research? What experience or training did the researchers have?

- Describe the method of data extraction for textual analysis (e.g., piloted forms, independently, in duplicate) and any processes for obtaining and confirming data from investigators.

Results

- Figure 2 ends up repeating the data present in table 3. I suggest allocating frequencies and percentages in the figure.

- Tables 5 and 6 only contains quotations in English, not in Chinese (p.6 line 221).

Discussion

- The discussion section should be reorganized because they are poor. I believe there should be better integration of the results with the existing literature. Discuss the generalisability (external validity) of the study results. Also, discuss both the direction and magnitude of any potential bias.

- Listing the limitations of the study.

- The implications for practice/research/education/policy should have been approached in greater depth. 

CHECKLIST FOR STYLE.

- The manuscript needs to be carefully and attentively proofread, because some sentences are awkwardly constructed, punctuation is deficient, and therefore reading is occasionally difficult to follow. Would recommend a thorough technical edit of this paper.

Round 2

Reviewer 2 Report

I appreciate the willingness to improve the study, as it has been done. The analysis is now more objective and neutral in the excellent sense of research work. By the other hand,  it shows better the qualities and advantages of the text mining method.

Reviewer 3 Report

I believe that the review carried out has greatly improved the quality of the study. Also, I do think that the author(s) addresses the broad questions, appropriately which were asked. Congratulations!